# The Emerging Role of Tumor Microenvironmental Stimuli in Regulating Metabolic Rewiring of Liver Cancer Stem Cells

**DOI:** 10.3390/cancers15010005

**Published:** 2022-12-20

**Authors:** Margherita Correnti, Eleonora Binatti, Elena Gammella, Pietro Invernizzi, Stefania Recalcati

**Affiliations:** 1Department of Biomedical Sciences for Health, University of Milan, 20133 Milano, Italy; 2Division of Gastroenterology, Center for Autoimmune Liver Diseases, Department of Medicine and Surgery, University of Milano Bicocca, 20900 Monza, Italy; 3European Reference Network on Hepatological Diseases (ERN RARE-LIVER), San Gerardo Hospital, 20900 Monza, Italy

**Keywords:** cancer stem cells, heterogeneity, plasticity, metabolism, liver cancer, tumor microenvironment

## Abstract

**Simple Summary:**

Increasing evidence indicates that cancer stem cells (CSCs) are the main factor responsible for therapeutic resistance and tumor relapse in primary liver cancer (PLC). This could also be related to the peculiar metabolic phenotype of liver CSCs. Although metabolic reprogramming is currently considered a hallmark of cancer, including PLC, no consensus has been reached with respect to the metabolic features of liver CSCs, which may shift between different metabolic states. CSC plasticity can be controlled not only by several cell-intrinsic cues but also by numerous cell-extrinsic stimuli derived from the tumor microenvironment (TME) surrounding the CSCs. In this review, we summarize the most significant discoveries regarding the metabolism of liver CSCs and highlight how the TME plays a role in regulating liver CSC metabolic plasticity. An improved understanding of the specific mechanism underlying liver CSC metabolic rewiring, as well as metabolic liver CSC–TME bidirectional crosstalk, is needed to develop valid new combinatorial therapeutic strategies.

**Abstract:**

Primary liver cancer (PLC) is one of the most devastating cancers worldwide. Extensive phenotypical and functional heterogeneity is a cardinal hallmark of cancer, including PLC, and is related to the cancer stem cell (CSC) concept. CSCs are responsible for tumor growth, progression, relapse and resistance to conventional therapies. Metabolic reprogramming represents an emerging hallmark of cancer. Cancer cells, including CSCs, are very plastic and possess the dynamic ability to constantly shift between different metabolic states depending on various intrinsic and extrinsic stimuli, therefore amplifying the complexity of understanding tumor heterogeneity. Besides the well-known Warburg effect, several other metabolic pathways including lipids and iron metabolism are altered in PLC. An increasing number of studies supports the role of the surrounding tumor microenvironment (TME) in the metabolic control of liver CSCs. In this review, we discuss the complex metabolic rewiring affecting liver cancer cells and, in particular, liver CSCs. Moreover, we highlight the role of TME cellular and noncellular components in regulating liver CSC metabolic plasticity. Deciphering the specific mechanisms regulating liver CSC–TME metabolic interplay could be very helpful with respect to the development of more effective and innovative combinatorial therapies for PLC treatment.

## 1. Introduction

Primary liver cancer (PLC) is the fifth most common cancer worldwide and the second leading cause of cancer-related mortality. Its incidence is rapidly increasing, especially in Western countries. Primary liver tumors are grossly classified as hepatocellular carcinoma (HCC) and cholangiocarcinoma (CCA), accounting for approximately 90% and 5% of all PLC, respectively [1,2,3].

Extensive phenotypical and functional heterogeneity is a hallmark of human tumors, including PLC, and represents one of the major contributing factors for tumor progression and the failure of therapeutic treatment. Tumor heterogeneity is classified as intertumor heterogeneity (tumors evolve differently over time between patients in terms of clonal structure, genotype and phenotype) and intratumor heterogeneity (different subpopulations of cancer cells coexist within the same tumor) [4].

Oncometabolism is considered an emerging hallmark of tumor heterogeneity. Dynamic changes in the bioenergetics machinery of cancers have been observed not only in different tumor types but also within the same tumor by creating specialized metabolic compartments. Moreover, modifications occurring in the tumor microenvironment (TME), such as changes in pH, hypoxia and lack of nutrients, further increase the level of complexity of this scenario. Although metabolic reprogramming is currently considered a hallmark of cancer [5,6], no consensus has been reached on the metabolic features of cancer stem cells (CSCs), especially those of liver CSCs.

The aim of this review is to elucidate the current knowledge on liver CSCs, focusing on the metabolic rearrangement that occurs in this peculiar tumor subpopulation. Furthermore, we provide a general overview of the influence of the TME on the metabolic plasticity of CSCs, as well as innovative and promising therapies targeting CSC metabolism, which could represent a pivotal strategy to improve liver cancer management.

## 2. CSC and Intratumor Heterogeneity 

Increasing evidence supports intratumor heterogeneity as a cardinal hallmark of cancer complexity. Two different models can explain intratumor heterogeneity: the clonal evolution model, which proposes intratumor heterogeneity as a consequence of natural selection, and the more recent CSC model, which states that intratumor heterogeneity depends on the ability of CSCs to differentiate into an heterogeneous tumor cell progeny (Figure 1) [7,8]. The clonal evolution model, also called the stochastic model, suggests that several mutation events generate tumor heterogeneity and progression and that every malignant cell may undergo genetic and/or epigenetic alterations and clonally expand to initiate tumor growth [9]. Genetic and epigenetic variations that occur over time in individual tumor cells can confer a selective heritable advantage in a Darwinian-like manner, allowing individual clones to survive and originate other clones that acquire additional advantageous mutations, consequently leading to heterogeneity among tumor cells within a single patient [4]. According to this model, tumor organization is not necessarily hierarchical, and from the clinical point of view, the rational way to cure and shrink cancer is to eliminate most or all tumor cells [10].

In contrast, the hierarchical or CSC model proposed by Bonnet and Dick may explain intratumor heterogeneity, proposing that only CSCs, with their unlimited self-renewal ability, can initiate and sustain tumor growth [9]. CSCs, or tumor-initiating cells (TICs), represent a portion of tumor cells endowed with the unique ability to self-renew and differentiate into “non-stem” cells, representing the driving force for tumor maintenance and progression, as well as for tumor relapse and resistance to conventional anticancer therapies [11,12,13,14]. Studies in mouse models of brain, intestinal and skin tumors support the crucial role of CSCs in tumorigenesis and clearly revealed that CSCs are essential for tumor repopulation and survival. Therefore, tumor eradication would require the killing of CSCs [15]. This model assumes that the unidirectional hierarchy of CSC division is responsible for tumor growth, i.e., only CSCs can generate the bulk of the tumor via symmetric division (for self-renewal) or asymmetric division (to generate differentiated cells).

The CSC model is considered an oversimplified description of tumor generation. Interestingly, several studies have shown that tumor hierarchy is not strictly unidirectional but more dynamic and fluid. In this respect, a third modern and more accepted theory, the ‘CSC plasticity model’, was recently proposed (Figure 1). According to this theory, cancer cells are very plastic and possess the dynamic ability to constantly shift between non-CSC and CSC states depending on various intrinsic and extrinsic stimuli, amplifying the complexity of understanding tumor heterogeneity [15,16,17,18].

The existence of CSCs has been confirmed in many solid tumors to date, including in HCC [19,20,21,22] and, recently, in CCA [23,24]; however, no consensus has been reached regarding the origin of putative liver CSCs. Classically, it has always been thought that CSCs originate from normal liver stem cells, but the idea that CSCs may originate from more committed progenitor cells or even from mature differentiated tumor cells through a reprogramming or dedifferentiation process has become widespread [3,18,25,26,27]. It is important to underline that the hierarchical or CSC plasticity model does not address the PLC cell-of-origin, which represents the cell acquiring the first cancer-promoting mutation(s) and is not necessarily related to the CSC concept [9].

## 3. Liver Tumor Metabolic Reprogramming and CSCs

Starting from the pioneering work of Otto Warburg, the concept of tumor metabolic reprogramming is of particular interest. In fact, it represents a recently emerging hallmark of cancer; several observations indicate that tumor genetic alterations also imply metabolic rearrangement [5,6,28]. Cancer cells display unique metabolic and bioenergetic features and can obtain energy from different substrates (e.g., glucose, lactate, pyruvate, fatty acids, glutamine, acetate) depending on various factors, such as the degree of vascularization/oxygenation of the tumor area, microenvironmental stress, etc. [29,30]. Highly proliferative cancer cells need to reprogram their metabolism to provide regular support for their increased division rate and rapidly generate increased amounts of ATP and synthesize more macromolecules compared to normal proliferating cells.

### 3.1. Glucose Metabolism

In a normoxic environment, somatic mammalian cells obtain energy and ATP through the tricarboxylic acid (TCA) cycle and mitochondrial oxidative phosphorylation (OXPHOS), namely cellular respiration or the electron transport chain, which is more efficient than the glycolytic pathway in terms of generating energy. In the cytoplasm of normal cells, glycolysis anaerobically converts one molecule of glucose to two molecules of pyruvic acid, releasing energy that is conserved as ATP and simultaneously reducing nicotinamide adenine dinucleotide (NAD^+^) to nicotinamide adenine dinucleotide phosphate (NADH) [28,31,32]. Specifically, one molecule of glucose yields two molecules of ATP, two molecules of NADH and two molecules of pyruvate. In the presence of oxygen, the two molecules of pyruvate are transported in the matrix of the mitochondria and transformed into two molecules of acetyl-CoA by the pyruvate dehydrogenase enzyme complex. Acetyl-CoA fuels the TCA cycle, which in turns produces NADH. Finally, NADH enters the electron transport chain in the aerobic phase, generating ATP more efficiently. At the end of this entire metabolic process, 38 molecules of ATP per mole of glucose are produced: 2 during glycolysis and 36 during OXPHOS [28,31,32]. Thus, the presence of oxygen is a crucial factor for the normal production of ATP in the cell. Conversely, in the absence of oxygen, the pyruvate derived from the anaerobic phase is converted into lactate, which is released in the bloodstream through a specific transporter, namely monocarboxylic acid transporter (MCT) [28,31,32].

Otto Warburg first described a peculiar phenomenon of cancer cells that switched the energy metabolism from the more efficient OXPHOS to the inefficient process of glycolysis, even in the presence of abundant oxygen supply, mainly because the ATP produced by glycolysis is 100 times faster than that produced by OXPHOS [33,34]. This change in cellular glucose metabolism is a typical biochemical hallmark of cancer cells. Warburg proposed that tumor metabolic reprogramming is a consequence of a permanent loss of mitochondrial OXPHOS functions. This hypothesis was rejected, as the impairment of OXPHOS has not been found in most common spontaneous tumors. This phenomenon can be explained as a result of the combination of various factors, such as the activation of oncogenes (e.g., c-Myc and Ras), the loss of tumor suppressors (e.g., p53 and PTEN), the presence of a hypoxic microenvironment and the occurrence of mitochondrial DNA mutations [35]. Interestingly, hexokinase 2 (HK2), the enzyme that catalyzes the first step of glucose metabolism (the phosphorylation of glucose to glucose-6-phosphate), is a transcriptional target of hypoxia-inducible factor 1 (HIF1), the pleiotropic factor that regulates tumor hypoxia [28,31,32] and plays critical roles in tumors shifting from OXPHOS to glycolysis [36]. High expression of HK2 was found in HCC tissues [37]. HIF1 also induces the expression of glucose transporters 1 (GLUT1) and 3 (GLUT3) and inhibits the conversion of pyruvate in acetyl-CoA by activating pyruvate dehydrogenase kinase 1 (PDK1) [36]. Increased expression of GLUT1 is observed not only in HCC [38] but also in CCA, where it correlates with poor tumor differentiation grade and with a significantly shorter overall survival of patients [39,40]; PDK1, PDK2 and PDK3 were also found to be significantly overexpressed in CCA tissues [41]. Recently, Raggi et al. confirmed the dependence of CCA cells on glycolysis to meet energy demands [42]. Moreover, increased levels of lactate, the final product of Warburg metabolism, have been observed in CCA, possibly depending on the observed overexpression of lactate dehydrogenase A (LDH-A), the enzyme that catalyzes the conversion of pyruvate to lactate, which is also associated with shorter survival in CCA patients [43,44]. In HCC cells, lactate seems to enhance ferroptosis resistance, contributing to tumor growth [45].

The pentose phosphate pathway (PPP) is a pathway parallel to glycolysis that uses glucose-6-phosphate to produce ribose-5-phosphate and NADPH, playing a critical role in the maintenance of redox balance [38]. PPP stimulation can be related to an increased antioxidant capacity and drug resistance in CCA cells [46]. Similarly, glucose-6-phosphate dehydrogenase (G6PD) and transketolase (TKT), two key enzymes of PPP, are upregulated and significantly associated with poor prognosis in HCC [47,48]. All this evidence highlights the importance of the shift of cancer cells from mitochondrial metabolism to glycolytic flux in PLC.

Unlike cancer cells, CSCs have heterogeneous metabolic characteristics, as they can utilize both glycolysis and OXPHOS. Interestingly, CSCs have been described as primarily glycolytic or preferentially relying on OXPHOS in a tumor-type-dependent manner; however, contradictory results for the same tumor entity have also been reported [49]. Fekir et al. reported angiopoietin-like 4 (ANGPTL4)-mediated changes in mitochondrial activity of HCC CSCs, with concomitantly reduced mitochondrial membrane potential and enhanced lactate production. These alterations were, in part, due to the ANGPTL4-induced overexpression of PDK4 accompanied by upregulation of glucose transporters (e.g., GLUT 3, 5 and 6) and glycolytic enzymes (e.g., phosphofructokinase platelet (PFKP) and pyruvate kinase M (PKM)), as well as downregulation of numerous genes involved in TCA and OXPHOS (e.g., cytochrome c oxidase (COX), NADH: ubiquinone oxidoreductase (NDUF), succinate dehydrogenase (SDH) and ubiquinol cytochrome c oxidoreductase chain (UQRC) families) [26]. Recently, Chen et al. confirmed the reprogramming of glycolytic metabolism in HCC CSCs, which showed an overexpression of glycolysis-related genes (e.g., genes encoding for GLUT1, HK2, PFK liver type (PFKL) and LDH-A), together with a downregulation of OXPHOS-related genes (e.g., cytochrome B (CytB), as well as ATP synthase membrane subunit 6 and 8 (ATP 6 and 8) genes]. Moreover, the authors demonstrated that this metabolic shift toward glycolysis is triggered by the expression of HBV x protein (HBx) [50]. The glycolytic switch of HCC CSCs was further demonstrated in other studies [51,52,53]. Moreover, an extensive transcriptomic and metabolomic characterization of CD133^+^ HCC CSCs revealed the central role of the oncogene Myc in their rearrangement of glycolytic metabolism [54]. Fan et al. reported a concomitant enhancement of the glycolytic pathway and mitochondrial biogenesis in HCC CSCs, an effect that was further strengthened by the expression of K+ channel protein potassium calcium-activated channel subfamily N member 4 (KCNN4) [55]. Recently, Raggi et al. shed light on CCA CSC metabolism, although the research on this topic remains very limited. With this work, the authors demonstrated enhanced mitochondrial respiration of CCA CSCs isolated by 3D sphere-forming assay. The observed changes in the CSC energy metabolic profile are likely related to the potentiated respiratory machinery, as demonstrated by the increased mitochondrial membrane potential and mitochondrial mass found in CCA spheres. Parallelly, CCA spheres showed downregulation of the glycolytic pathway with reduced expression of glycolysis-related genes (GLUT1, HK2 and PKM2) and reduced glucose uptake, together with minor production of lactate [42]. In contrast, Tamai et al. reported a BEX2-dependent suppression of mitochondrial activity in dormant CD274^low^ CCA CSCs, a mechanism dependent on the interaction between BAX2 and the mitochondrial protein tu translation elongation factor (mitochondrial TUFM) [56].

Hence, CSCs show a peculiar metabolic phenotype that can be glycolytically or OXPHOS-oriented. Whereas some studies have demonstrated that quiescent CSCs are more glycolytic than differentiated tumor cells, other reports suggest that CSCs are less glycolytic and have increased mitochondrial mass and membrane potential, preferring mitochondrial oxidative metabolism. However, in either case, mitochondrial function seems to be critical to maintain CSC features (stemness, drug resistance and invasion ability).

Moreover, the overexpression of peroxisome proliferator-activated receptor gamma coactivator 1-alpha (PGC1α) seems to be responsible for the boosted mitochondrial metabolism of metastatic tumor cells displaying CSC features [31]. Interestingly, the central role of PGC1α as a master regulator of mitochondrial biogenesis in the CSC subset has been confirmed in CCA. The pharmacological inhibition of PGC1α (with the use of SR-18292) or knockdown in CCA CSCs was reported to not only impair the mitochondrial mass but also reduce CCA stem-like features [42].

### 3.2. Lipid Metabolism

Glucose metabolism alone cannot account for CSC metabolism, and the different CSC metabolic pathways are very flexible and closely intertwined with each other. In fact, alterations in lipid- and cholesterol-associated pathways are essential for CSC maintenance, overall in order to meet their bioenergetics needs. Cancer cells can use fatty acids (FAs) as an energy source to sustain their proliferation, survival and invasion [28,38]. FAs can be derived through bloodstream uptake or, alternatively, via abnormally activated de novo lipid synthesis. In the first case, free FAs can be actively intracellularly transported through the action of transmembrane transporters such as CD36, and this transportation may be facilitated by binding with specialized small fatty acid binding proteins, such as fatty acid binding proteins 1 and 5 (FABP1, FABP5) [38]. All these FA transportation machineries are upregulated in HCC. In particular, overexpression of FABP1 also promotes angiogenesis and metastasis of HCC [57], whereas the hyperactivation of the FABP5/HIF1α axis is associated with a high recurrence rate and poor prognosis among HCC patients [58]. CD36 upregulation has also been found in CCA tissues [28], whereas FABP5 overexpression seems to characterize only the extrahepatic form of CCA [59,60]. Moreover, the absorption of lipoproteins, another source of exogenous lipids, is increased in HCC, mainly thanks to the upregulation of lipid lipase (LPL), the overexpression of which correlates with poor prognosis of patients [61,62]. Ruiz de Gauna et al. demonstrated an augmented uptake of very-low-density and high-density lipoproteins (VLDL and HDL) in the extrahepatic CCA cell line EGI1, resulting in the accumulation of neutral lipids, such as triglycerides and cholesteryl ester content [60]. Furthermore, exogenous accumulated lipids can be used as a source of energy by tumor cells through fatty acid β-oxidation (FAO). In line with the high content of FAs, several enzymes with a role in the FAO pathway are dysregulated in HCC, such as the rate-limiting enzyme carnitine palmitoyl transferase 1 (CPT1) and other lipolysis-related enzymes that are upregulated in HCC, as well as medium/long-chain acyl-CoA dehydrogenase (MCAD/LCAD), which is downregulated in HCC [38]. The FAO rate is also increased in highly proliferative extrahepatic CCA EGI1 cells, which are characterized by increased levels of acyl-CoA dehydrogenase medium chain (ACADM), a protein involved in the first reaction of FAO [60]. An alternative way to obtain FAs is to reactivate de novo lipid synthesis, as previously mentioned. In this case, cancer cells synthetize new FAs using cytoplasmatic acetyl-CoA as a substrate. In HCC, enhanced lipogenesis is enabled by the increase in several lipogenic enzymes in terms of both activity and expression, such as ATP citrate synthase (ACLY) (converts citrate in acetyl-CoA and oxaloacetate, in addition to enhancing glycolysis), acetyl-CoA carboxylase (ACC) (converts acetyl-CoA into malonyl-CoA), fatty acid synthase (FASN) (forms palmitate and other fatty acid products starting from malonyl-CoA and acetyl-CoA), acyl-CoA synthase (ACS) (converts FAs to fatty acyl-CoA esters) and stearoyl-CoA desaturase enzyme 1 (SCD1) (converts saturated FAs to monounsaturated FAs) [63,64,65,66,67,68]. Unlike HCC, CCA is characterized by the downregulation of FASN, suggesting an independence of this form of PLC in terms of de novo FA synthesis, as demonstrated by in vitro and in vivo evidence [60,69]. Cholesterol metabolism appears to be very important for highly proliferative tumor cells, as demonstrated by the deregulation of the transcription factors sterol regulatory element-binding protein 1 (SREBP-1), the upregulation of which promotes HCC growth and metastasis [70], and liver X receptor (LXR), which regulates both cholesterol homeostasis and lipogenesis and was found to be downregulated in HCC tissues [38,71].

With respect to the CSC compartment, it has been demonstrated that CSCs highly rely on FAO to fulfill their energy demands under glucose-limiting conditions [9,72]. The upregulation of several stemness markers (e.g., EPCAM, ITGA6, CD133 and CD44) in the EGI1 cell line, which simultaneously shows increased FAO, supports this hypothesis in the context of CCA. Then, increased FAO rate in turn supports the increased mitochondrial oxidation of CCA CSCs [60]. ChIP-seq analysis performed by Chen et al. revealed NANOG as the connecting point between FAO and CSC features in HCC. This likely depends on the simultaneous OXPHOS repression and FAO activation actions exerted by the transcription factor. In fact, through the binding with specific gene promoters, NANOG repressed the transcription of OXPHOS genes and, conversely, stimulated the transcription of those involved in the FAO pathway, ultimately favoring HCC CSC self-renewal and chemoresistance. Interestingly, NANOG induces an upregulation of SCD1, the enzyme involved in the rate-limiting step in the formation of monounsaturated FAs, suggesting that lipid desaturation is also required for NANOG-mediated CSC generation [73]. The role of SCD1 expression in regulating liver CSCs has been also supported by other studies reporting its negative correlation with HCC tumor differentiation grade and its association with disease progression and poor clinical outcomes [67,74,75]. In contrast, Bort et al. reported a decrease in FAO in HCC CSCs. Results obtained by the authors indicate enhanced lipid accumulation as cytoplasmic lipid droplets in sorafenib-induced HCC CSCs, likely due to both extracellular lipid uptake and de novo lipogenesis, together with a reduction in β oxidation. The AKT-SREBP1-dependent upregulation of ACLY, ACC and FASN, together with an enhanced expression of CD36, was reported. Downregulation of PGC1α, PPARγ and carnitine palmitoyl transferase 1 (CPT-1) was also demonstrated, which is indicative of a β-oxidation inhibition. This work pointed out the importance of another pair of signaling pathways for lipid metabolism reprogramming of HCC CSCs, namely the phosphoinositide 3-kinase and AKT (PI3K–AKT) pathways [76].

### 3.3. Iron Metabolism

Iron is necessary for several key biochemical processes, such as cellular replication and growth, and is thus an essential element for life. However, both iron deficiency and iron excess have the potential to be harmful. The former can cause cell growth arrest and death, whereas excess iron may facilitate the formation of the most reactive and toxic forms of oxidants through Fenton reactions [77]. Thus, iron homeostasis is strictly controlled. The major players in the regulation of cellular iron homeostasis are the transferrin receptor (TfR1), which is responsible for transferrin-bound iron uptake; ferroportin (Fpn), the only known cellular iron exporter; and ferritin (Ft), an iron storage protein. Specifically, the complex formed by transferrin-bound Fe^+3^ and TfR1 is internalized via endocytosis. Then, the acidic environment of the endosome causes the release of Fe^+3^ and its subsequent reduction to ferrous iron (Fe^+2^), which is delivered to the cytoplasm, where it is made available for cellular biological processes (labile iron pool or LIP). Excess iron can be stored as Ft reserves or extracellularly released through the Fpn [78]. Moreover, iron regulatory proteins 1 and 2 (IRP1 and IRP2) regulate iron balance at the post-transcriptional level [79]. Interestingly, the abundance of IRP2 is controlled in an iron-dependent manner by iron-sensing F-box and leucine-rich repeat protein 5 (FBXL5), ablation of which in an FBXL5-deficient mouse model leads to IRP2 overactivity-dependent iron overload, promoting oxidative stress, lipid peroxidation, tissue damage, inflammation and compensatory proliferation of hepatocytes, resulting in the promotion of HCC development in the case of exposure to diethylnitrosamine in mice. Results obtained in FBXL5-deficient mice been then validated in HCC patient samples from five different cohorts, confirming the involvement of the FBXL5-IRP2 axis in liver carcinogenesis [80]. Furthermore, this and many other studies underline the close connection between iron overload and PLC development [81,82]. Indeed, hepatic iron accumulation is associated with HCC development in patients with hereditary haemochromatosis (HH), a genetic disorder characterized by excessive absorption of iron in many organs, including the liver, as well as with nonalcoholic steatohepatitis-related cirrhosis [83,84,85]. Interestingly, the association between iron overload and HCC development was also confirmed in the absence of fibrosis and cirrhosis, suggesting a direct role of hepatic iron overload in sustaining the malignant transformation of hepatocytes [86]. As previously mentioned, systemic iron deficiency is associated with the progression of several cancer types, including PLC. In fact, the expression of proteins involved in iron uptake, storage and export are often altered in cancer cells, which generally present with increased iron uptake and decreased iron release, resulting in major iron sequestration and an elevated LIP in tumor cells, with consequent improper systemic iron levels available for erythroblasts. All these processes can eventually give rise to functional iron deficiency anemia, a condition associated with liver tumors [87,88,89,90].

Hence, iron can contribute to both tumor initiation and progression, and the current model of altered iron homeostasis in cancer cells can be summarized within a framework in which increased iron uptake and decreased iron storage and export contribute to enhanced cellular iron levels in order to sustain the requirements of fast-growing tumor cells [82]. Increased iron uptake by HCC cells has been confirmed by many investigations. For example, Shen et al. used data from *The Cancer Genome Atlas* (TCGA) to analyze gene expression levels of iron-related genes in 373 HCC samples compared to 50 non-cancerous adjacent tissues and from the *Human Protein Atlas* to obtain immunohistochemical staining of iron-related proteins in normal and liver cancer tissues [91]. The analysis performed by the authors demonstrated an increased expression of TfR1, Ft heavy chain (FtH) and feline leukemia virus subgroup C receptor 1 (FLVCR1) in HCC tissues at the both gene and protein levels. Moreover, the investigation of the prognostic value of some iron metabolism-related genes revealed an association of the expression of genes encoded with TfR1 (TFRC) and Ft light chain (FtL) with a shortened overall survival of HCC patients, whereas the expression of FLVCR1 was associated with both overall survival and tumor recurrence. Examining the clinicopathological data of HCC samples, the authors also linked the increased expression of TFRC with high vascular invasion and tumor histological grade, and they reported a correlation between FLVCR1-augmented expression and higher tumor stage, tissue inflammation, vascular invasion and tumor histological grade, suggesting the potential role of TfR1 and FLVCR1 in HCC pathogenesis [91]. The relevance of enhanced TFRC expression in hepatocarcinogenesis was also confirmed by Sun et al., who reported elevated TFRC levels in Hepa1-6 tumor-bearing mice. Moreover, the authors demonstrated that the increased iron absorption by HCC cells restricted iron uptake by tumor-associated macrophages, leading to their polarization towards a tumor-growth-promoting immunosuppressive phenotype. These results were then confirmed in human patients using gene expression profiles and clinical data of HCC samples included in the TCGA database [92]. Several studies support a role of iron in CCA patients. High iron accumulation was also observed in CCA and correlated with poor prognosis of patients. Specifically, TfR1 expression was significantly upregulated in malignant cholangiocytes compared to normal bile ducts and correlated with CCA metastasis. Increased expression of LIP and TfR1 was also confirmed in two patient-derived CCA cell lines [93]. TfR1 overexpression in CCA tissues was also confirmed by Sarcognato et al., who additionally reported increased levels of glutathione peroxidase 4 (GPX4), the expression of which was related to poor prognostic histological parameters. Moreover, given the role of GPX4 in inhibiting ferroptosis, an association between its expression and reduced survival of CCA patients was also found [94]. Raggi et al. recently showed an association of shorter survival with higher iron levels in isolated CCA tumor cells [95]. Additionally, Mancinelli et al. make a distinction between the small- and large-duct types of intrahepatic CCA (iCCA). In particular, Fpn expression was found to be increased in small-duct-type iCCA, whereas TfR1 and Ft were upregulated in both iCCA subtypes, although to a limited extent in large-duct-type iCCA. Hepcidin levels were also elevated in every type of iCCA; a possible explanation is related to the dependence of its activation on interleukin-6 (IL-6), which is commonly upregulated in CCA [96].

Different iron-driven mechanisms could induce PLC development, among which the most important is undoubtedly the generation of reactive oxygen species (ROS), as previously mentioned. Excess in ROS production could, in turn, promote liver carcinogenesis, triggering hydroxyl and lipid radical productions, mutagenesis and genomic instability [86]. Other iron-driven mechanisms include the promotion of cell proliferation and the downregulation of tumor suppressor p53 [97]. However, the precise mechanism of action of iron overload in inducing hepatocarcinogenesis remains elusive.

It is largely accepted that augmented iron content is a key feature of CSCs in several tumor types and that dysregulated iron metabolism is functionally required for CSC maintenance, as demonstrated by higher in vivo tumorigenic ability of iron-rich 3D tumor spheres [18,98,99]. Moreover, altered expression of iron-related proteins in CSCs is commonly associated with a poor prognosis in many human tumors [95,100,101,102,103]. Although the connection between altered iron homeostasis and PLC is well known, investigation of iron metabolism rearrangement in the liver CSC compartment is still in initial stages in comparison with other tumor systems; therefore, additional research is warranted. Recently, Raggi and colleagues demonstrated that increased iron content characterized CCA CSCs, as demonstrated by a higher amount of FtH and lower levels of Fpn in CCA 3D tumor spheres compared to parental cells growing as a monolayer. According to these findings, CCA spheres were characterized by augmented expression of heme oxygenase (HO-1) and ROS levels, both of which are indicative of elevated oxidative stress. These results are accompanied by important changes in the expression of stemness-related genes when iron content is manipulated via exposure of parental monolayer cells and 3D spheres to ferric ammonium citrate (FAC) or to iron chelator desferrioxamine (DFO), respectively. The authors also analyzed the transcriptomes of 104 CCA patients, comparing the expression of major iron-related genes in tumor (T) and matched surrounding liver (SL) tissues. Interestingly, they found a downregulation of FtH and Fpn genes, together with an upregulation of TfR1, in T compared to SL, in line with the iron expression pattern characterizing several other tumors. Next, a correlation of FtH and hepcidin gene expression with a trend towards shorter survival was shown in CCA patients [95]. Additionally, Xiao et al. underlined the key role of TfR1 in the regulation of stemness features of HCC CSCs. Their results demonstrated a strong upregulation of TfR1 in CSCs derived from two different HCC cell lines, with a consequential iron accumulation. Moreover, highly malignant behaviors were promoted by TfR1 expression, as revealed by the knockdown of TfR1 mRNA levels. TfR1 downregulation blocked the cell cycle in the G1/G0 phase, inhibited invasive capacity, decreased the colony formation ability and reduced the ROS levels of HCC CSCs. Results also indicated that TfR1 expression was essential in sensitizing 3D tumor spheres to erastin-induced ferroptosis, as well as n maintaining the stemness features of CSCs [104].

## 4. Effect of TME on Liver CSC Metabolic Plasticity

As demonstrated by all the previous evidence, CSCs are characterized by a marked metabolic flexibility and may shift between different metabolic phenotypes depending on TME-derived stimuli. It is well known that CSC plasticity can be controlled by several cell-intrinsic and cell-extrinsic cues, the latter of which are derived from the complex microenvironment around the CSCs, the so-called CSC niche (Figure 2) [9].

### 4.1. Intrinsic Factors Controlling CSC Plasticity

Intrinsic factors include genetic and epigenetic perturbations. Several studies underline the importance of genetic changes, especially transcription factors, in the regulation of cellular plasticity [105,106]. For example, Oct4, c-Myc, NANOG and Sox9 are linked to CSC plasticity in liver cancer [107]. Moreover, pluripotency-related transcription factors, such as NANOG, Wnt or Oct4, are strictly interlaced with some regulators of CSC metabolism, such as c-Myc, p53 and HIF1α, inextricably linking stemness properties with metabolic rearrangement. It is important to underline that metabolic regulators themselves can also sustain CSC stemness properties [108,109]. Although genetic alterations play a prominent role in regulating CSC plasticity, they are not sufficient alone and need to be accompanied by a variety of epigenetic alterations, including changes in DNA methylation, chromatin remodeling, histone modification patterns and non-coding RNA (ncRNA) activity [15,17,110]. In liver cancer, histone deacetylase sirtuin-1 (SIRT1) is crucial for the maintenance of self-renewal in liver CSCs. In fact, SIRT1 is overexpressed in liver CSCs, and its inhibition by short hairpin RNA (shRNA) significantly suppress the in vitro sphere-forming efficiency and in vivo tumorigenic potential of HCC CSCs. These effects are likely due to the SIRT1-mediated epigenetic regulation of Sox2 expression [111]. SIRT1 can also act by deacetylating β-catenin in liver CSCs, maintaining its stability and promoting its nuclear accumulation, with a consequent overexpression of NANOG, which is a key regulator of liver CSC stemness and self-renewal [112]. Moreover, Raggi et al. showed that DNA methyltransferase 1 (DNMT1) inhibition-driven epigenetic reprogramming determines the formation of a liver CSC pool by long-lasting cell-context-dependent memory effects [21]. The role of ncRNAs as key regulators of several CSC biological processes, such as asymmetric division and unresponsiveness to treatments, has recently emerged [17]. Importantly, aberrantly genetically activated pathways that drive tumorigenesis could also reprogram tumor cell metabolic networks, which can, in turn, influence the activity of different regulatory enzymes involved in chromatin remodeling. For example, acetyl-CoA represents the substrate of some enzymes with histone acetylation activity. Gain-of-function mutations in isocitrate dehydrogenases 1 and 2 (IDH1 and IDH2) frequently associated with CCA development can also modulate the activity of α-ketoglutarate-dependent dioxygenases, which are responsible for a variety of post-translational epigenetic modifications, such as CpG island hypermethylation [32].

### 4.2. Extrinsic Cues Regulating CSC Plasticity: Effect of Non-cellular and Cellular Components of the TME

In addition to genetic and epigenetic alterations, extrinsic factors linked to the CSC niche in terms of surrounding stromal cells, the extracellular matrix (ECM) and the release of several mediators (summarized in Table 1) play a crucial role in influencing CSC plasticity [113,114,115,116,117]. The HCC/CCA TME consists of heterogeneous cell populations (e.g., liver CSCs, more differentiated tumor cells, infiltrating immune cells and other stromal cells) and non-cellular components (e.g., ECM, hypoxia, nutrient-deprived conditions, several released cytokines, growth factors, metabolites, etc.), which differentially contribute to the regulation of CSC plasticity in general and CSC metabolic reprogramming in particular [118].

#### 4.2.1. Role of TME Noncellular Components in Controlling CSC Metabolic Plasticity

CSCs can show different metabolic phenotypes to adapt to different TME conditions, in particular oxygen tension and glucose concentrations. Hypoxia, a common feature in solid malignancies, is characterized by low oxygen, acid stress and nutrient deprivation. It develops as consequence of the rapid growth of the tumor, which outstrips the oxygen supply, as well as the generation of abnormal blood vessels transport [119]. Under hypoxic conditions, only selected subpopulations of cells can survive, and it has been widely reported that hypoxic stress specifically supports CSC maintenance and reprogramming. HIF1α, the principal mediator of hypoxic effects on tumor cells, regulates CSC stemness, modulating various signaling pathways, including the Notch, Hedgehog, Hippo, wingless (Wnt)/β-catenin, Janus-activated kinase/signal transducer, activator of transcription (JAK/STAT), phosphatidylinositol 3-kinase/phosphatase, tensin homolog (PI3K/PTEN) and nuclear factor-kB (NF-kB) pathways [119]. Hypoxia is also known to promote CSC stemness through the activation of HIF1α in liver CSCs [37,120]. In fact, hypoxia can regulate the expression of CSC marker CD44, as well as the as the activation of Oct4 transcription factor. Moreover, the NF-kB/HIF1α pathway is critical to the maintenance of the stemness of CD133^+^ HCC CSCs [118]. In CCA, hypoxia causes the aberrant activation of the Sonic Hedgehog pathway, which is frequently deregulated in CCA CSCs, as well as the upregulation of some CSC transcription factors (e.g., NANOG, Oct4 and Sox2) and markers (e.g., CD133). Hypoxia also plays a pivotal role in CCA invasion, as demonstrated by enhanced expression of epithelial–mesenchymal transition (EMT)-related markers (e.g., N-cadherin and Vimentin) under hypoxic conditions [121]. Hypoxia is a well-known stimulus that can reprogram CSC metabolism towards a more glycolytic phenotype [122]. Ling et al. recently demonstrated the existence of a positive feedback loop between ubiquitin-specific protease 22 (USP22) and HIF1α in HCC cells upon p53 inactivation. Specifically, the hypoxia-mediated activation of HIF1α induced the upregulation of USP22, which in turn acted to deubiquitinate HIF1α, leading to the expression of HIF1α target genes, including glycolysis-related genes HK2, PDK1 and enolase 1 (ENO1) [123]. These findings further support the close relationship between stemness and metabolism in liver CSCs.

Apart from hypoxia, glucose deprivation is another common feature of the TME of solid tumors, including the PLC TME. As previously mentioned, under conditions of glucose deprivation, CSCs possess the ability to shift from aerobic glycolysis (generally preferred by CSC in a glucose-rich environment) to a mitochondrial oxidative metabolism for energy production [31]. Glucose deprivation promotes the expansion of the liver CSC compartment, as demonstrated by in vitro and in vivo experiments performed by Loong et al., who proposed a mechanism of action according to which restricted glucose availability activates the protein kinase RNA-like endoplasmic reticulum kinase (PERK)/Eukaryotic translation initiation factor 2 alpha (eIF2α)/activating transcription factor 4 (ATF4) signaling pathway, leading to the activation of fucosyltransferase 1 (FUT1) promoter activity. The final result of this cascade consists of the enhancement of liver CSC frequency, as well as an increasing in sorafenib resistance. Notably, the FUT1-dependent regulation of liver CSC stemness is mediated by the activation of the AKT/mechanistic target of rapamycin kinase (mTOR)/eukaryotic translation initiation factor 4E binding protein 1 (4EBP1) signaling axis [124].

It is important to underline that some stemness-associated transcription factors (e.g., NANOG, Wnt and Oct4) are inextricably interlaced with CSC metabolic regulators (e.g., c-Myc, p53 and HIF1α), strictly linking the stemness with metabolic rearrangement [108,109].

#### 4.2.2. Regulatory Effect of TME Cellular Components on Liver CSC Metabolic Features

As mentioned above, cellular components within the TME play a role in the regulation of CSC plasticity. Among these, tumor endothelial cells (TECs) are crucial players in the formation of abnormal neovessels (tumor angiogenesis) that supply tumors with oxygen and nutrients and provide a route of metastasis. Tumor blood vessels are characterized by unorganized patterns consisting of irregular monolayers of TECs and do not have an endothelial barrier function [125]. All these abnormalities result in vascular leakiness with a random blood flow (hypoxia). Compared to normal endothelial cells, TECs acquire a proangiogenic phenotype; are highly migratory, proliferative and self-sustaining; and release several cell-attracting factors (e.g., damage-associated molecular patterns (DAMPs)) for inflammatory immune cells [125]. TECs in the TME are more than simply components of blood vessels; they may influence CSC stemness by actively secreting various factors, as demonstrated in many tumor types, including PLC [98,126,127]. In vitro experiments with HUVEC-conditioned medium confirmed the contribution of endothelial cells to the regulation of liver CSC stemness, as demonstrated by the enhanced sphere-forming ability and increased expression of CSC marker CD133 [128]. On the other hand, liver CSCs release several angiogenic factors, the first of which is vascular endothelial growth factor (VEGF), thereby activating TECs and consequently promoting angiogenesis, as demonstrated by many studies [129,130,131].

Cancer-associated fibroblasts (CAFs) are a key stromal component that play a fundamental role in tumor initiation, growth, invasion and dissemination and represent the major component of tumor stroma in both HCC and CCA [98,132]. Importantly, bidirectional crosstalk occurs between CAFs and cancer cells. CAFs are chronically activated by tumor-derived factors and can, in turn, provide metabolic support to tumor cells [133]. Increasing evidence shows that CAFs also support stemness features of PLC. Li et al. demonstrated that the stemness-promoting action of CAFs was related to their secretion of IL-6 and hepatocyte growth factor (HGF), which acted by activating STAT3 signaling in CD24+ HCC CSCs [134]. Moreover, another study performed by Liu et al. confirmed that CAFs are involved in stemness maintenance of liver CSCs by activating the Notch3 pathway, leading to lysine-specific demethylase 1 (LSD1) stability [135]. Recently, Lin et al. elucidated the mechanism through which CAFs promote CCA stemness. Briefly, CCA CAFs released IL-6 and IL-33, which stimulated the hyperactivation of 5-lipoxygenase (5-LO) metabolism in recruited myeloid-derived suppressor cells (MDSC), potentiating their stemness-enhancing ability. The hyperactivation of 5-LO metabolite signaling in CAF-educated MDSC led to the overproduction of the metabolite leukotriene B4 (LTB4), which bound its receptor, leukotriene B4 receptor type 2 (BLT2), to CCA cells, ultimately promoting CCA stemness via the PI3K/Akt/mTOR signaling pathway [136]. CAFs also play a role in sustaining the excessive metabolic needs of tumor cells, including CSCs, mainly by the secretion of metabolites such as lactate and pyruvate, which can be absorbed by tumor cells and utilized as an alternative carbon source [137]. CAFs function to reprogram the metabolism of neighboring cancer cells and CSCs through the secretion of various signaling molecules [138]. Direct evidence of bidirectional metabolic interaction between CAFs and tumor cells was recently described as a “reverse” Warburg effect, a process by which tumor cells cause a metabolic reprogramming of surrounding CAFs that switch to a more glycolytic phenotype, with the consequent production of high levels of lactate. Then, lactate is exported through MCT4 and recaptured by tumor cells via the MCT1 transporter. In turn, cancer cells may convert the lactate into pyruvate, fueling OXPHOS metabolism [28,138].

CSC metabolic plasticity can also be triggered by a variety of signals emanating from immune cells. Crosstalk between inflammation and cancer development has been found in many solid tumors, including PLC [139,140,141]. It is well known that PLC is frequently associated with a chronic inflammation status characterized by the infiltration of several immune cell types. Notably, this persistent inflammatory microenvironment is crucial for CSC regulation, as CSC stemness features can be regulated by the cytokines and factors released by immune cells, such as tumor necrosis factor α (TNFα), IL-6 and interferon-γ (IFN-γ) [118]. Specifically, inflammatory cytokines TNFα and IL-1β activate NF-kB, consequently upregulating transcription of IL-6. In turn, IL-6 triggers the JAK2/STAT3 cascade, which plays a critical role in liver CSCs expansion [142]. Furthermore, in vitro and in vivo treatment with IFN-γ enrich the fraction of CD133^+^ liver CSCs [143]. Among infiltrated immune cells, tumor-associated macrophages (TAMs) are predominant components, and their strict interaction with CSCs is well known. In HCC, TAMs reinforce and maintain CSC stemness through the release of different factors, including IL-6, transforming growth factor beta (TGF-β) and TNFα, which induce the activation of several signaling pathways involved in stemness maintenance (e.g., STAT3, Wnt/β-catenin, Notch, NANOG) [118,144]. M2-derived TNFα also promotes cancer stemness through the Wnt/β-catenin pathway [145]. TAM-derived IL-6 triggers the expansion of CD44^+^ HCC CSCs by activating the STAT3 signaling pathway [116]. Recently, Wei et al. reported the role of another inflammatory protein secreted by TAMs, S100A9, in the regulation of the self-renewal capacity of HCC CSCs [146]. In addition, tumor-associated neutrophils (TANs) can interact with liver CSCs, controlling the progression of PLC. For example, TAN-derived bone morphogenetic protein 2 (BMP2) and TGF-β2 induce the expression of miRNA (miR)-301-3p in HCC cells, with a subsequent enhancement of stem-like features [147]. Accumulating evidence indicates that the activation of some stemness-related pathways (e.g., NF-kB and PI3K/Akt) by inflammatory cytokines (e.g., IL-6 and IL-8) not only stimulates CSC self-renewal and maintenance but also triggers CSC metabolic rearrangement (e.g., NF-kB and PI3K/Akt-dependent induction of glycolysis), strengthening the connection between inflammation and the metabolic plasticity of CSCs.

**Table 1 cancers-15-00005-t001:** TME-derived factors that play a role in the regulation of liver CSC plasticity.

Mediators	Secretory Cells	Receptor Cells	Refs
DAMPS	TECs	Immune cells	[125]
Angiocrin factors		Liver CSCs	[97,126,127,128]
IL-6, HGF	CAFs	CD24+ HCC CSCs	[134]
IL-6, IL-33		MDSCs	[136]
Lactate, Piruvate		Liver CSCs	[137]
LTB4	MDSCs	CCA CSCs	[136]
TNFα, IL-1β, IFN-γ	Inflammatory Cells	Liver CSCs	[115]
IL-6, TGF-β, TNFα	TAMs	HCC CSCs	[113,115,144,145]
S100A9		Liver CSCs	[146]
BMP2, TGF-β2	TANs	HCC CSCs	[147]

## 5. Conclusions 

In recent decades, there has been an increase in awareness of the role of oncometabolism as an emerging hallmark of cancer progression, providing a major new boost to research in the oncology field. Thanks to several studies on this topic, the general mechanisms controlling tumor metabolic rearrangement have been defined in many types of tumors, including PLC, albeit to a limited extent (especially in CCA). It is now becoming clear that these alterations in energy metabolism also affect the CSC compartment and are inextricably intertwined with stemness features of tumors.

To date, very little is known about the specific mechanism regulating liver CSC metabolic rewiring. In this view, it is imperative to more clearly decipher the metabolic heterogeneity and plasticity of liver CSCs in order to develop novel therapeutic strategies targeting the CSC-TME metabolic interplay, which could be used in combination with traditional standard therapies to induce increased sensitivity of CSCs. Moreover, multiple obstacles must be overcome to effectively eliminate CSCs, the first of which is the identification of a specific unique liver CSC marker. Another unresolved issue is the elucidation of the metabolic differences between CSCs and non-CSCs, as well as between CSCs and normal progenitor stem cells, which is crucial to selectively target the CSC compartment. More detailed studies on these issues are needed.

### Future Directions: Targeting CSC Metabolism as a Promising Therapeutic Opportunity

Metabolic targeting of CSCs could represent an important emerging field of cancer therapies that can effectively eliminate CSCs, as the main factor responsible for chemo-resistance and tumor relapse. This is of particular importance for PLC, given that its unresponsiveness to conventional therapy remains one of the major challenges in liver cancer research. It has been shown that high glycolytic activity in CSCs is interlinked with their stemness properties. Thus, targeting glycolysis could be a promising therapeutic approach to reduce this aggressive cancer subpopulation [148]. Glucose metabolism could be targeted using different inhibitors of glycolytic enzymes, such as the 2-deoxy-D-glucose (2-DG), which inhibits the activity of HK2, or 3-bromopyruvic acid (3-BrPA), which not only inhibits HK2 but also glyceraldehyde-3-phosphate dehydrogenase (GAPDH), phosphoglycerate kinase 1 (PGK1) and LDH [38]. However investigation of glycolysis inhibitors, such as the glucose transporter inhibitors WZB117, fasentin, phloretin, and silybin/silibinin, has shown little success, mainly owing to the ubiquitous expression of glycolytic enzymes [149]. Another potential target is the OXPHOS dependency of CSCs. Accordingly, it has been demonstrated that metformin can exert a marked cytotoxic effect on CSCs by inhibiting the electron transport chain complex I [9]. For instance, the use of metformin in HCC induces an intense decrease in the proportion of EpCAM+ CSCs [150]. Moreover, CCA CSCs also exhibit evident sensitivity to metformin, not only reducing the survival of CCA 3D spheres but also decreasing the expression of genes related to stemness, self-renewal, pluripotency, drug resistance and EMT [42]. Hence, targeting mitochondrial OXPHOS could be a promising therapeutic strategy. It has to be underlined that because CSCs cover their energy needs mainly through OXPHOS and glycolysis, simultaneous targeting of these two pathways could be an alternative and more effective therapeutic approach for their eradication [148]. Another potential strategy to eradicate CSCs relies on blocking essential enzymes and factors involved in fatty acid synthesis, such as ACC1 or ACC2 (e.g., by ND-654), FASN (e.g., by TVB-26409), ACLY and SCD1 (e.g., by A939572) [38]. On the contrary, inhibition of FAO by etomoxir decreases the fraction of CD133^+^ HCC CSCs, as well as resistance to sorafenib [73]. Additionally, targeting transcriptional factors such LXR could be relevant (e.g., using the LXR agonist TO901317) [71]. Considering the importance of iron for CSC formation and stemness maintenance, modifying the iron metabolism could be helpful in order to suppress the growth of CSCs. Both iron chelation and iron overload have been proposed for cancer treatment. One of the most used iron chelators is desferrioxamine, although it has demonstrated limited success, owing to its poor availability. A new class of iron chelators including thiosemicarbazone Dp44m appears to be very promising, although they have not yet been tested on CSCs [18]. Alternatively, triggering ferroptosis, a nonapoptotic form of regulated cell death involving iron-dependent lipid peroxidation, seems to be promising to attack CSCs. However, a recent study demonstrated that CCA CSCs are more resistant to ferroptosis inducer erastin, so enhancing intracellular iron levels is not always the appropriate strategy to eliminate CSCs [18,99]. It has also been shown that iron-sensing ubiquitin ligase FBXL5, a previously unrecognized oncosuppressor in liver carcinogenesis in mice, and dysregulation of FBXL5-mediated cellular iron homeostasis are associated with poor prognosis in human HCC, suggesting that FBXL5 plays a key role in defense against hepatocarcinogenesis. Therefore, FBXE can also be considered a potential therapeutic target for HCC associated with cellular iron dysregulation [80].

Importantly, researchers have also begun to elucidate the role of TME as a key and active participant not only in stemness maintenance but also in the plethora of metabolic changes involving CSCs. For this reason, targeting TME-derived stimuli may represent a new antitumor strategy. An example is represented by the HIF1α inhibitory approach with the use of HIF1α inhibitors (e.g., RO7070179, EZN-2968 and ENMD-1198) [118]. On the other hand, it could also be useful to modulate the tumor-infiltrating stromal cells and their secreted molecules. In the context of HCC, targeting CAF-derived IL-6 using an IL-6 neutralizing antibody has achieved promising results in terms of EMT prevention and modulation of stem-like features [151,152]. Furthermore, targeting TAM-derived secretome might be helpful to improve the efficacy of conventional therapies. In this context, the antagonistic relationship between the metabolic sensor SIRT1, a class III histone deacetylase with strong expression in metabolic tissues such as the liver, and transcription factor NF-κB, a master regulator of inflammatory response, in controlling inflammation must also be considered. For this reason, SIRT1 targeting is emerging as a potential strategy to improve different metabolic and/or inflammatory pathologies, as well as to prevent tumor initiation and progression [153].

Despite all the evidence reported above, the metabolic rewiring of liver CSCs and the direct liver CSCs-TME stimuli metabolic interplay are still largely unexplored, necessitating further investigations, especially in the field of CCA. Given the importance of the interplay between CSCs and their TME, researchers are now developing new antitumor strategies to target CSC niche factors and disrupt the communication between CSCs and the TME.

It is also essential to note that CSCs do not merely respond to the TME but can actively create their own immunosuppressive niche through the secretion of several soluble mediators and factors. The final result is immune evasion, a common hallmark of CSCs [118,154]. For example, HCC CSCs can actively recruit macrophages and drive their polarization towards an immunosuppressive M2 phenotype by secreting C-C motif chemokine ligand 2 (CCL2), macrophage colony-stimulating factor (M-CSF) and IL-8 [130,155]. Moreover, chemoresistant HCC cells, particularly those enriched in the CSC component, activate the expansion and the immunosuppressive functions of myeloid-derived suppressor cells through the release of IL-6 [156]. The secretion of CCL22 by CD44^+^ HCC CSCs also plays a role in the creation of an immunosuppressive milieu, mediating the recruitment of regulatory T cells (Tregs) [157,158]. A recent study demonstrated that CSC-derived VEGF, rather than M-CSF, drives macrophage differentiation in CCA. Moreover, the secretion of IL-13, IL-34 and osteoactivin by CCA tumor spheres is implicated in the macrophage acquisition of a CSC-specific phenotype characterized by the concomitant expression of M1 and M2 transcription traits and the presence of tumor-promoting functions [24]. Moreover, the nutrient deprivation state of the TME and the accumulation of metabolites contribute to triggering an immunosuppressive phenotype of immune-infiltrating cells. For example, tumor-derived extracellular lactate accumulation increases the proportion of Tregs, which, in turn, reduces the IFN-γ secretion of effector T cells [159]. In addition, lactate promotes the polarization of macrophages to an M2 state and stimulates angiogenesis (e.g., by inducing VEGF secretion) [32]. Thus, another possibility to disrupt CSC-TME crosstalk could rely on targeting all CSC-derived factors implicated in TME re-education, such as CCL2, IL-8, CCL22, VEGF, IL-13, IL-34 and osteoactivin, with specific inhibitors.

## Figures and Tables

**Figure 1 cancers-15-00005-f001:**
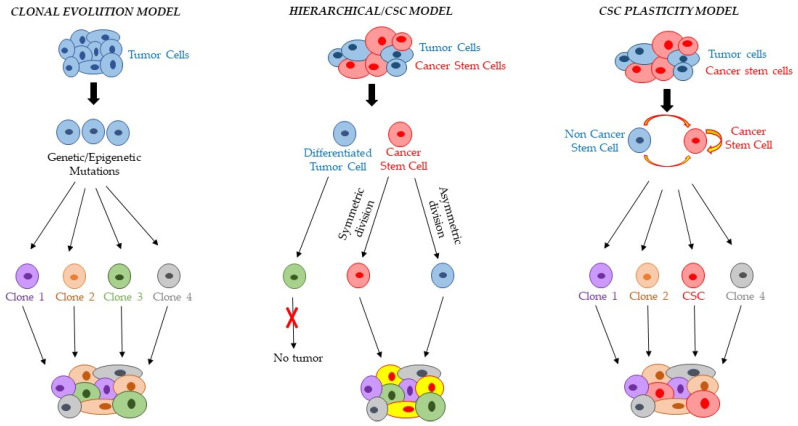
**Schematic overview of different models that can explain intratumor heterogeneity.** According to the clonal evolution model, all cells in the tumor have equal tumorigenic potential. The tumor cells may undergo genetic and/or epigenetic alterations, generating different tumor clones with the same ability to initiate tumor growth. On the contrary, the hierarchical or CSC model suggests that only cancer stem cells (CSCs) can initiate and sustain tumor growth. CSCs can generate heterogeneous tumor cell progeny via asymmetric division and maintain the CSC component via symmetric division. The CSC plasticity model assumes that tumor hierarchy is very dynamic. Plastic cancer cells can constantly shift between non-CSC and CSC states depending on various intrinsic and extrinsic stimuli, giving rise to a heterogeneous tumor population.

**Figure 2 cancers-15-00005-f002:**
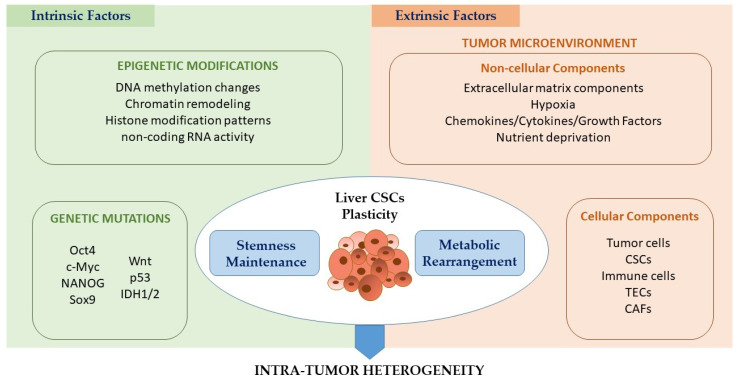
**Schematic representation of intrinsic and extrinsic cues that contribute to CSC plasticity.** CSC plasticity can be controlled by several cell-intrinsic and cell-extrinsic cues. Intrinsic factors include genetic mutations of various transcription factors/genes (e.g., Oct4, c-Myc, NANOG, Sox9, Wnt, p53, IDH1/IDH2) and epigenetic variations (e.g., DNA methylation changes, chromatin remodeling, histone pattern modification and non-coding RNA activity). On the other hand, the extrinsic factors are related to the surrounding tumor microenvironment (TME), which consists of heterogeneous cell populations (e.g., CSCs, more differentiated tumor cells, infiltrating immune cells and other stromal cells) and non-cellular components (e.g., ECM, hypoxia, nutrient-deprived conditions, several released cytokines, growth factors, metabolites, etc.0d). All these factors differentially fuel the plasticity of liver CSCs, regulating stemness maintenance and the metabolic rearrangement of liver CSCs.

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
