# Peer review of "The Emerging Role of Tumor Microenvironmental Stimuli in Regulating Metabolic Rewiring of Liver Cancer Stem Cells"

_cancers, 2022, doi:10.3390/cancers15010005_

Round 1
Reviewer 1 Report
The manuscript presented by Correnti et al reviews the metabolic reprogramming of liver cancer stem cells and the influence of the tumor microenvironment, both in hepatocellular carcinoma (HCC) and in cholangiocarcinoma (CCA). The authors have collected interesting evidence on this topic and have condensed this knowledge well in this review.
However some issues should corrected in the manuscript submitted.
- Take care of providing good quality images. According to this version for revision, the quality of the figures should be improved, particularly in Figure 1.
- Regarding the results obtained in FBXL5-335 deficient mice (line 338), please provide reference.
- Regarding the crucial role of SIRT1 in liver CSC please specify a reference about it and a comment.
- Identify the Sheen et al manuscript mentioned in line 357 (reference 90?).
- If considered of interest, please Introduce in the discussion the following references:
Muto et al. Disruption of FBXL5-mediated cellular iron homeostasis promotes liver carcinogenesis. J Exp Med 2019.
De Gregorio et al. Relevance of SIRT1-NF-κB Axis as Therapeutic Target to Ameliorate Inflammation in Liver Disease. Int J Mol Sci. 2020
Papadaki et al. Regulation of Metabolic Plasticity in Cancer Stem Cells and Implications in Cancer Therapy. Cancers 2022.
Verify along the manuscript:
- Additional spaces in the text: e.g. Page 1 line 27, Page 2 line 45, Page 3 line 96 and 100, Page 4 line 139, Page 5 line 198 and 208…
- Consistent text size: e.g. Page 2 line 45-47.
- Check for missing characters and typos: e.g. Page 10 line 449, Page 11 line 468, several in Table 1 (DAMPS, Secretory) …
Author Response
We thank this reviewer for appreciating our manuscript and providing helpful comments.
The manuscript presented by Correnti et al reviews the metabolic reprogramming of liver cancer stem cells and the influence of the tumor microenvironment, both in hepatocellular carcinoma (HCC) and in cholangiocarcinoma (CCA). The authors have collected interesting evidence on this topic and have condensed this knowledge well in this review. However some issues should corrected in the manuscript submitted.
- Take care of providing good quality images. According to this version for revision, the quality of the figures should be improved, particularly in Figure 1.
- Regarding the results obtained in FBXL5-335 deficient mice (line 338), please provide reference.
- Regarding the crucial role of SIRT1 in liver CSC please specify a reference about it and a comment.
The text has been changed as suggested.
- Identify the Sheen et al manuscript mentioned in line 357 (reference 90?).
This reference (91) is right and it refers to entire paragraph (Line 357-372). Now to simplify we introduce the ref also at the beginning of the sentence.
- If considered of interest, please Introduce in the discussion the following references: Muto et al. Disruption of FBXL5-mediated cellular iron homeostasis promotes liver carcinogenesis. J Exp Med 2019.De Gregorio et al. Relevance of SIRT1-NF-κB Axis as Therapeutic Target to Ameliorate Inflammation in Liver Disease. Int J Mol Sci. 2020Papadaki et al. Regulation of Metabolic Plasticity in Cancer Stem Cells and Implications in Cancer Therapy. Cancers 2022.
We introduce in the discussion the references suggested.
Verify along the manuscript:
- Additional spaces in the text: e.g. Page 1 line 27, Page 2 line 45, Page 3 line 96 and 100, Page 4 line 139, Page 5 line 198 and 208…
- Consistent text size: e.g. Page 2 line 45-47.
- Check for missing characters and typos: e.g. Page 10 line 449, Page 11 line 468, several in Table 1 (DAMPS, Secretory) …
The text has been changed as suggested.
Reviewer 2 Report
the manuscript is nice, good work done. I accept it in its present form, just separately describe the future direction before conclusion
1. The main question addressed by the research is very much clear. this question can be raised as it's worthful.
2. The conclusion should be to the point and self-explanatory and recommended
3. References are okay, Tables and figures are okay
Author Response
Thank you for your appreciation of our work and for helpful suggestions.
The manuscript is nice, good work done. I accept it in its present form, just separately describe the future direction before conclusion
- The main question addressed by the research is very much clear. this question can be raised as it's worthful.
- The conclusion should be to the point and self-explanatory and recommended
- References are okay, Tables and figures are okay
We separated conclusion and future direction